Prolonged decay of molecular rate estimates for metazoan mitochondrial DNA

Molak Martyna 1 2 martyna.molak@gmail.com
Ho Simon Y.W. 1
1 School of Biological Sciences, University of Sydney , Sydney , Australia
2 Museum and Institute of Zoology, Polish Academy of Sciences , Warsaw , Poland
Amos William
Electronic publication date: 2015 Mar 5
Publication date: 2015
Volume: 3
Electronic Location ID: e821
Received 2014 Nov 28; Accepted 2015 Feb 16
Copyright: © 2015 Molak and Ho
Copyright year: 2015
Copyright holder: Molak and Ho
License: This is an open access article distributed under the terms of the Creative Commons Attribution License, which permits unrestricted use, distribution, reproduction and adaptation in any medium and for any purpose provided that it is properly attributed. For attribution, the original author(s), title, publication source (PeerJ) and either DOI or URL of the article must be cited.
License URL: https://creativecommons.org/licenses/by/4.0/

Keywords: Substitution rate, Molecular clock, Evolutionary timescale, Calibration, Time-dependent rates

Funding: University of Sydney International Scholarship Australian Research Council Martyna Molak was supported by the University of Sydney International Scholarship. Simon Y.W. Ho was supported by the Australian Research Council. The funders had no role in study design, data collection and analysis, decision to publish, or preparation of the manuscript.

==============================
Evolutionary timescales can be estimated from genetic data using the molecular clock, often calibrated by fossil or geological evidence. However, estimates of molecular rates in mitochondrial DNA appear to scale negatively with the age of the clock calibration. Although such a pattern has been observed in a limited range of data sets, it has not been studied on a large scale in metazoans. In addition, there is uncertainty over the temporal extent of the time-dependent pattern in rate estimates. Here we present a meta-analysis of 239 rate estimates from metazoans, representing a range of timescales and taxonomic groups. We found evidence of time-dependent rates in both coding and non-coding mitochondrial markers, in every group of animals that we studied. The negative relationship between the estimated rate and time persisted across a much wider range of calibration times than previously suggested. This indicates that, over long time frames, purifying selection gives way to mutational saturation as the main driver of time-dependent biases in rate estimates. The results of our study stress the importance of accounting for time-dependent biases in estimating mitochondrial rates regardless of the timescale over which they are inferred.

Introduction

Understanding the tempo and mode of the molecular evolutionary process is one of the fundamental goals of biological research. Determining rates and timescales of evolution allows us to explore such questions as the co-evolution of species (Reed et al., 2007), causes of extinction (Lorenzen et al., 2011), and drivers of diversification (Jetz et al., 2012). Molecular estimates of evolutionary timescales are made using the molecular clock, which assumes that DNA evolves at a constant rate among lineages (Zuckerkandl & Pauling, 1962). With improving knowledge of the evolutionary process, molecular-clock methods have undergone considerable development over the past five decades (Ho, 2014; Kumar, 2005), with various forms of rate variation being able to be taken into account (Ho & Duchêne, 2014).

To estimate evolutionary rates from DNA sequence data, molecular clocks need to be calibrated. This involves using independent information to constrain the age of one or more nodes in a phylogenetic analysis. There are several types of evidence that can provide calibrations for molecular clocks (for a recent review, see Hipsley & Müller, 2014). One of the primary sources is the fossil record, which can provide an indication of the first appearance of a lineage and thus place a minimum bound on when it diverged from its sister lineage. The ages of nodes in the tree can also be estimated using biogeographic hypotheses. For example, the rise of barriers to gene flow or new habitats for colonization can provide age estimates for divergences between sister species (e.g., Fleischer, McIntosh & Tarr, 1998).

Over short timescales, molecular clocks can be calibrated by the ages of heterochronous samples or by available pedigree records. Heterochronous sampling is common in studies of viruses and of ancient DNA. If the ages of the samples are sufficiently distinct from one another, they can be used to calibrate estimates of evolutionary rates (Drummond et al., 2003; Molak et al., 2013). In pedigree studies, the relationships among samples have been documented and the divergence times might be exactly known.

The most significant challenge to the molecular clock has been widespread evidence of rate variation among lineages. These patterns of rate variation can be partly explained by differences in generation time, longevity, and other life-history characters (Bromham, 2009). Recently, however, there has been growing evidence of heterogeneity in rate estimates across timescales. For example, mitochondrial rates estimated in phylogenetic studies are about an order of magnitude lower than those estimated in analyses of pedigrees (Ho & Larson, 2006; Howell et al., 2003). More generally, estimated rates appear to be time-dependent, showing a negative relationship with the timescale over which they are measured (Ho et al., 2011; Ho et al., 2005). This pattern has been observed in coding and non-coding mitochondrial DNA from a range of taxa, including humans (Endicott et al., 2009; Henn et al., 2009; Rieux et al., 2014), fish (Burridge et al., 2008; Genner et al., 2007), birds (García-Moreno et al., 2006; Ho et al., 2005), and invertebrates (Crandall et al., 2012; Gratton, Konopiński & Sbordoni, 2008; Ho & Lo, 2013; Papadopoulou, Anastasiou & Vogler, 2010).

Various explanations have been proposed for time-dependent biases in rate estimation. These include natural selection, calibration errors, model misspecification, sequence errors, and biases in phylogenetic estimation and sampling. The first three factors are likely to be the most important (see Ho et al., 2011 for a review). Transient deleterious mutations tend to inflate estimates of rates over short timescales, but they are removed by selection over longer timescales (Duchêne, Holmes & Ho, 2014; Soares et al., 2009; Woodhams, 2006). Calibration errors mostly derive from the assumption that the genetic split between lineages coincides with the divergence of populations. The impact of calibration error is more severe in short-term rate estimates, where the discrepancy between the timing of genetic and population splits constitutes a large proportion of the overall timescale (Ho et al., 2005; Peterson & Masel, 2009). Additionally, estimates of rates can be misled if they are obtained using models that provide a poor reflection of real evolutionary and demographic processes (Emerson, 2007; Navascues & Emerson, 2009; Soubrier et al., 2012). Misspecified models of nucleotide substitution can lead to underestimates of saturation, causing underestimates of rates over long evolutionary timescales (dos Reis & Yang, 2013; Duchêne, Holmes & Ho, 2014; Ho et al., 2007).

The time-dependent biases in molecular rate estimation have important implications for studies of recent evolutionary timescales. For example, the ages of coalescence events within populations will potentially be overestimated if they are based on long-term rates calibrated using fossil data (Ho et al., 2008). This time-dependence, together with substantial variation in rates among different taxa, also calls into question the application of so-called ‘standard’ mitochondrial rates, such as those estimated in studies of lizards (Macey et al., 1998), birds (Shields & Wilson, 1987; Weir & Schluter, 2008), mammals (Brown, George & Wilson, 1979), and arthropods (Brower, 1994) and used in many subsequent studies. Time-dependent biases in rate estimation might also contribute to a wider discussion about discrepancies between molecular and palaeontological estimates of divergence times (Benton & Ayala, 2003).

Although there is a clear discrepancy between rates estimated on short and long timescales, there remains uncertainty about how rates scale with time. A major hindrance has been the paucity of reliable calibrations on intermediate timescales. In this regard, ancient DNA data can be particularly useful in bridging the gap between short- and long-term rate estimates (Ho, Kolokotronis & Allaby, 2007). Because ages are assigned directly to sequences in the analysis, no assumptions need to be made about the timing of population splits and their correspondence to genetic divergences (Rambaut, 2000). Unfortunately, ancient DNA analyses are constrained by the post-mortem degradation of DNA and the limited reach of radiocarbon dating.

Accurate characterization and quantification of time-dependent biases in rate estimation is crucial for understanding their causes and accounting for their effects. Studies of time-dependence in rate estimates have typically involved limited data sets, comprising small numbers of taxa or sequences. This leaves open questions concerning the prevalence and ubiquity of these biases, the timescales across which rate estimates are time-dependent, and the rate of decay between high pedigree rates and low phylogenetic rate estimates.

Quantifying the extent of time-dependent rates is particularly important because it can provide insights into its causes. For example, purifying selection is only expected to operate over relatively short timescales, whereas mutational saturation tends to become a problem at greater time depths (Duchêne, Holmes & Ho, 2014; Subramanian et al., 2009). On the other hand, if saturation has only a small impact on rate estimates, there should be an upper limit to the timescale over which the biases can be observed. If these limits and the mode of decay of rate estimates can be identified and incorporated into phylogenetic models, the accuracy of the inference of evolutionary processes will be substantially improved.

In this study, we aim to characterize the taxonomic breadth and temporal depth of time-dependent rates by performing a meta-analysis of mitochondrial rates estimated from a wide range of amniote and insect taxa. We investigate how these rate estimates depend on the age of the calibrations used to estimate them, how they vary among taxonomic groups and molecular markers, and whether time-dependent biases persist across a broad temporal scale.

Methods

Collection of published rate estimates

We performed a meta-analysis of evolutionary rate estimates to determine their relationship with the timescale over which they are estimated. Our data set comprised published estimates of molecular evolutionary rates, along with the ages of their corresponding calibration points (Table S1). We collected rate estimates for amniotes and insects from recent publications in several leading journals in molecular evolution (Table S1).

Because methods used to estimate rates varied widely among studies, we chose the oldest calibration used in each study as a proxy for the timescale over which the rate had been estimated. Therefore, for each rate estimate, ‘calibration time’ represents (i) the oldest fossil or biogeographical event used as an age constraint on a basal node for the phylogeny; (ii) the oldest ancient DNA sequence used in heterochronous sampling; or (iii) the depth of the genealogy in a pedigree study.

Novel rate estimates

In addition to collecting a set of published rate estimates, we obtained thirty novel estimates and re-estimates of rates by analysing available ancient DNA sequences from 16 species (Table S2). Twelve of these estimates were obtained using ancient DNA as the sole source of calibration. The remaining 18 estimates were obtained from ancient DNA data sets, but with the addition of a single sequence from a sister species; for these data sets a fossil-based estimate of the interspecific divergence time was used to calibrate the rate estimate. Identification of sister species and divergence times was done with reference to published studies (Table S2). One additional rate estimate was obtained using a mitochondrial data set from modern humans and an orangutan, calibrated using fossil evidence of the divergence time between these two species (Table S2).

Table 1 Linear-regression analysis of data points from different time slices using a sliding window with a width of four orders of magnitude, or with a shrinking window for the oldest calibrations.

Significant regression results (p ≤ 0.05) are shown in bold. A more detailed version of this table is available as Table S4.

		Calibration ages	
		100−104
years	101−105
years	102−106
years	103−107
years	104−108
years	105−109
years	106−109
years	107−109
years	108−109
years	
Coding markers	N	1	9	20	132	176	173	162	50	5	
R 2	0	0.05	0.24	0.31	0.23	0.09	0.04	0.11	0.04	
P-value	n/a	5.67 × 10−1	2.67×10 −2	3.43×10 −12	1.48×10 −11	4.79×10 −5	1.26×10 −2	1.76×10 −2	7.6 × 10−1	
Slope	n/a	−0.33	−0.34	−0.37	−0.29	−0.20	−0.15	−0.34	0.36	
Non-SSC a	n/a	–	***	***	***	***	***	***	–	
Non-coding markers	N	8	25	27	43	48	31	29	12	1	
R 2	0.12	0.27	0.33	0.62	0.55	0.27	0.17	0.16	0	
P-value	4.09 × 10−1	7.41×10 −3	1.80×10 −3	4.47×10 −10	1.41×10 −9	2.58×10 −3	2.55×10 −2	2.00 × 10−1	n/a	
Slope	0.08	−0.34	−0.36	−0.48	−0.49	−0.58	−0.53	−0.89	n/a	
Non-SSC a	**	***	***	***	***	*	*	–	n/a	
Notes.

a Results of tests against spurious self-correlation (non-SSC). Higher p-value from the two tests is shown.

* p < 0.05.

** p < 0.005.

*** p < 0.0005.

The best-fitting model of nucleotide substitution was chosen for each data set according to the Bayesian Information Criterion using ModelGenerator 0.85 (Keane et al., 2006). Substitution rates were estimated using the Bayesian phylogenetic software BEAST 1.7.2 (Drummond & Rambaut, 2007). We used a strict clock calibrated only by the ages of the sequences in the analyses of intraspecific data (Brown & Yang, 2011; Drummond et al., 2006) and an uncorrelated lognormal relaxed clock when analysing the data sets that included sister species. A uniform prior of 0–10−4 substitutions/site/year was specified for the mean substitution rate, whereas a 1/x prior was used for the population size in the coalescent prior for the tree. In analyses of data sets that included sister species, an informative prior distribution was specified for the age of the root according to the fossil-based estimate of the divergence time (Table S2). Posterior distributions of parameters were estimated using Markov chain Monte Carlo sampling, with samples drawn every 103 steps over a total of at least 107 steps. Some chains were extended to ensure sufficient sampling and effective sample sizes above 100 for all parameters.

For each data set the analysis was repeated using a Bayesian skyride model with a gamma prior for the precision parameter. The fit of the two population models (constant size and skyride) were compared using Bayes factors calculated in Tracer v1.5 (Rambaut & Drummond, 2007). We chose the rate estimate obtained using the better-fitting model. For all further analyses, we used median estimates of the strict-clock rate from the intraspecific analyses and the mean tree-wide rate from the analyses that included sister species.

Meta-analysis of the rate estimates

We conducted a meta-analysis of all of the rate estimates (previously published and generated in this study) by plotting log-transformed rate estimates against log-transformed calibration times. To avoid biasing the regression analysis with multiple estimates for single species, estimates from the various human pedigree studies were pooled together and used as a single data point (Table S1). For the same reason, we took the average of the human rate estimates calibrated using the Homo-Pan divergence time (Table S1). When a study reported rate estimates from a number of mitochondrial markers for a single species, we took the average of the estimates.

We did separate linear regressions for coding and non-coding markers, and for subsets based on taxonomic classes (insects, reptiles, birds, and mammals). We also performed an intraspecific regression on the human data to test for time-dependent rates at the intraspecific level (Table S1). All linear regressions were performed in R (R Development Core Team, 2014).

To test whether the strength of time-dependence varied across timescales, we performed regression analyses for subsets of the data, chosen according to calibration times. We used a sliding window with a width of four orders of magnitude (100–104 years, 101–105 years, …105–109 years). To investigate the extent of the timescale over which time-dependent biases in rate estimation can be observed, we performed regression analyses on subsets of data based on calibrations of 106–109, 107–109, and 108–109 years.

Testing for spurious self-correlation

Any correlation between substitution rates and timescales over which they are estimated might be a mathematical artifact caused by self-correlation. This phenomenon can occur when a ratio is plotted against its denominator and can produce spurious correlations that are highly significant (Kenney, 1982). Here, the ratio is the estimate of the substitution rate (genetic distance divided by time elapsed) and the denominator is the time elapsed.

For all of our regression analyses, we investigated whether the correlation we detect is solely such a mathematical artifact (Kenney, 1982; Sheets & Mitchell, 2001) using a randomization test (Jackson & Somers, 1991). For all of the data points in our analysis, we randomized the genetic distances, which we calculated by multiplying substitution rate and calibration time. We recalculated substitution rates using these randomized distances, then performed regression analyses against the corresponding calibration times. This procedure was repeated 10,000 times for each subset of data for which regression analysis was performed in our meta-analysis.

We then tested whether the estimate of the regression slope for the original data fell within the distribution of the estimated slopes for the randomized data. In this way, we determined whether or not the detected correlations were entirely the product of spurious self-correlation. Deviation from the distribution produced by the randomized data would signify that the observed time-dependence could not be exclusively a mathematical artifact.

For all data subsets tested, the randomized data produced an average regression slope of −1, which is the value expected when there is spurious self-correlation (Kenney, 1982). Therefore, we performed an additional test by repeating the regression analysis for each subset of data using a null hypothesis of slope = − 1.

Results

Our analyses were based on a data set comprising 239 published, re-estimated, and novel estimates of evolutionary rates in metazoan mitochondrial DNA (Table S1). Rate estimates ranged from 1.78 × 10−10 substitutions/site/year (cytochrome b and tRNA genes in woolly mammoth; Rohland et al., 2010) to 2.31 × 10−6 substitutions/site/year (the control region in Adélie penguins; Lambert et al., 2002). Calibration ages ranged from 6 years (pedigree study of Adélie penguins; Millar et al., 2008) to 4 × 108 years (fossil calibration for Bacillus stick insects; Plazzi, Ricci & Passamonti, 2011).

To test the relationship between mitochondrial rate estimates and calibration ages, we performed separate linear-regression analyses of coding and non-coding regions. All linear regressions were performed using log-transformed values of both estimated rates and calibration times. We found significant correlations between these two variables in coding markers (slope = −0.28, p = 1.58 × 10−12, R2 = 0.24) as well as non-coding markers (slope = −0.43, p = 1.38 × 10−12, R2 = 0.60) (Fig. 1 and Table S3).

Figure 1 Linear regressions of log-transformed rate estimates from mitochondrial markers in a range of metazoan taxa against the log-transformed calibration times that were used to estimate the rates.

Separate analyses were performed for coding markers (A) and non-coding markers (B).

To test whether time-dependent biases were present across the entire timescale that we studied, we performed additional regression analyses for subsets of the data that represented different calibration ages. We found significant non-zero relationships for all time slices containing more than 12 data points (Table 1 and Table S4). This includes time depths of >107 years for coding markers and >106 years for non-coding markers.

We found significant negative relationships between mitochondrial rate estimates and calibration ages in our separate analyses of coding and non-coding markers in mammals, birds, reptiles, and insects (Fig. 2 and Table S3). However, there were too few data points from non-coding markers for reptiles (2 data points) and insects (0 data points) to allow regression analysis. We also found support for time-dependent rates in our intraspecific analysis of mitochondrial sequence data from humans (Fig. S1 and Table S3), corroborating results previously obtained from a smaller data set (Henn et al., 2009).

Figure 2 Linear regressions of log-transformed rate estimates against log-transformed calibration times used for their estimation for different taxonomic groups (insects (A), reptiles (D), birds (B, E), and mammals (C, F)) and mitochondrial marker types (coding (A–D) and non-coding (E, F)).

There were insufficient data for regression analyses of non-coding markers in reptiles and insects.

Each of the significant regressions reported above was significantly different from a corresponding regression that was calculated after randomizing the data. Each was also significantly different from a null hypothesis of slope = −1, which was the average slope produced by randomized data (Table 1, Tables S3 and S4) as expected in the case of spurious self-correlation (Kenney, 1982).

Discussion

Our analyses demonstrate a consistent pattern of time-dependent biases in rate estimates in mitochondrial DNA across metazoan taxa. We observed this pattern across the entire timescale that was analysed, beyond at least 106 years in non-coding markers and 107 years in coding markers (Table 1 and Table S4). This is in contrast with previous estimates of the temporal depth of time-dependence, which has been variously estimated at a few hundred years (Richards M in Gibbons, 1998), around 50 kyr (Henn et al., 2009), 200 kyr (Burridge et al., 2008), or 1–2 Myr (Ho et al., 2005; Papadopoulou, Anastasiou & Vogler, 2010). However, our finding of a prolonged decay in molecular rate estimates is consistent with recent evidence from a large-scale analysis of substitution rates in viruses, which revealed a time-dependent bias in rate estimates across a temporal scale spanning 10 orders of magnitude (Duchêne, Holmes & Ho, 2014). Together, these results have considerable implications for studies of divergence times.

Our finding of a prolonged relationship between rate estimates and calibration times sheds some light on the causes of time-dependent biases in rate estimation. Specifically, the temporal breadth of the relationship between rate and calibration time suggests that more than one factor is responsible. The effects of purifying selection decline over time and thus are not expected to be detectable over long timescales (Nei, 1971; Phillips, 2009), even though the lack of recombination in mitochondrial DNA can potentially extend the life of deleterious mutations through genetic hitchhiking (Ballard & Kreitman, 1995; Birky & Walsh, 1988). In turn, mutational saturation is expected to increase over time and, if the available nucleotide substitution models do not correct for multiple hits sufficiently, the rates over longer timescales will be more severely underestimated. Underestimation of the number of substitutions across longer timeframes can lead to time-dependent patterns in other evolutionary parameters, including the ratio of nonsynonymous to synonymous substitutions (dos Reis & Yang, 2013) and the ratio of transitions to transversions (Duchêne, Ho & Holmes, in press). Therefore, we propose that purifying selection and mutational saturation are the main drivers of these biases. This is consistent with the patterns found in sequence data from viruses (Duchêne, Holmes & Ho, 2014).

Signs of a time-dependent bias in rate estimation persisting for more than 10 Myr and in a wide range of taxa also allow us to challenge some other hypotheses about the causes of time-dependent rates. Population fluctuations and other demographic factors (Balloux & Lehmann, 2012; Henn et al., 2009), which vary among species, cannot fully explain the patterns that we have observed across a diverse range of taxa. Given the temporal depth of time-dependent biases in rate estimation, calibration errors are unlikely to represent a sufficient explanation (Ho et al., 2011; Peterson & Masel, 2009). Our data-randomization analyses suggest that the time-dependent pattern of rate estimates is unlikely to be solely a mathematical artefact. Even if this were the case, however, accounting for time-dependent biases in rate estimates would still be necessary when estimating evolutionary timescales using the molecular clock.

Our results also support a gradual decay from high pedigree to low phylogenetic rate estimates through time. Intermediate rate estimates, such as those obtained from ancient DNA studies, have previously been attributed to various methodological biases (Debruyne & Poinar, 2009; Emerson & Hickerson, in press; Navascues & Emerson, 2009). However, our results do not indicate that rate estimates based on ancient DNA are anomalous. Additionally, estimates of rates calibrated using biogeographic events in the late Pleistocene and Holocene are comparable to those based on ancient DNA data of similar age. This can be seen, for example, in our intraspecific analysis of rate estimates from humans (Fig. S1 and Table S1).

One of the disadvantages of our meta-analysis is the paucity of mitochondrial rate estimates based on young calibrations for coding markers and for insects. Although the uneven representation of taxonomic classes on different timescales might affect the regression analysis that includes all data points, our analyses of class-specific subsets of data yielded significant evidence of time-dependent trends. One other potential source of bias in the meta-analysis is that studies of evolutionary processes on short timescales tend to focus on fast-evolving taxa, whereas studies of longer timescales have a greater focus on slowly evolving organisms (Sheets & Mitchell, 2001). This, however, can be rejected as a major cause for the biases based on the observed intraspecific time-dependent variation of estimated rates, which was shown previously (Duchêne, Holmes & Ho, 2014; Henn et al., 2009) and in this study (Fig. S1 and Table S3).

Our results confirm that considerable caution needs to be exercised in molecular studies of evolutionary timescales. The need to correct for time-dependent biases in rate estimates has been pointed out previously (Ho et al., 2005). As the variability of regression slopes in our study shows, however, there is no universal function that can be applied across taxa and across markers. Although a few studies have proposed corrections for specific taxa (Cutter, 2008; Gignoux, Henn & Mountain, 2011; Obbard et al., 2012; Soares et al., 2013; Soares et al., 2009), they focused on the effects of purifying selection. Additionally, it is not clear to what extent they can be extrapolated to other taxa. Ultimately, it might be best to use only calibrations that are reasonably close in time to the evolutionary events of interest (Ho & Larson, 2006).

In view of the prevalence of time-dependent biases in estimates of mitochondrial rates, neutrally evolving nuclear loci might be a preferable alternative to the mitochondrial genome for estimating evolutionary timescales. However, there has been compelling evidence of time-dependent biases in estimates of nuclear rates in some organisms, including Caenorhabditis elegans (Denver et al., 2004) and Drosophila melanogaster (Haag-Liautard et al., 2007). In contrast, studies of human nuclear genomes have yielded lower estimates of rates from pedigrees than from fossil-calibrated comparisons with chimpanzees (Roach et al., 2010; Xue et al., 2009). This represents a prominent anomaly against the broad time-dependent biases seen in other estimates of evolutionary rates, and it remains to be seen whether the results from human genomes are representative of a broader trend among nuclear genomes (Scally & Durbin, 2012).

Supplemental Information

Supplemental Information 1 Supplementary article

Intraspecific time-dependent rates—human case study; and Supplementary material references.

Click here for additional data file.

Figure S1 Linear regression of log-transformed rate estimates for human mitochondrial control region against the log-transformed calibration times that were used to estimate the rates.

Click here for additional data file.

Table S1 Published and novel rate estimates used in the study.

Click here for additional data file.

Table S2 Novel rate estimates and re-estimates obtained in this study. Analyses were done using either the ages of the ancient DNA sequences as calibrations, or by including the sister species and using fossil evidence to calibrate the divergence between the two species. Information about the used data set, sister species, details of fossil calibrations and the population model used in Bayesian phylogenetic inference are provided. Substitution rate estimate and calibration time for each data set were, along with previously published ones (Table S1), included in the meta-analysis. Cal, calibration.

Click here for additional data file.

Table S3 Linear-regression analysis of log-transformed rate estimates against the log-transformed calibration times that were used to estimate the rates for data subsets.

Click here for additional data file.

Table S4 Linear-regression analysis of log-transformed rate estimates against the log-transformed calibration times that were used to estimate the rates. Analyses were done for time-slice subsets of data using a sliding window with a width of four orders of magnitude or using a shrinking window for the oldest calibration times.

Click here for additional data file.

Supplemental Information 7 PRISMA flow diagram for the data collected and used in our meta analysis

Click here for additional data file.

The authors would like to thank William Amos, Brian Golding, and Mario dos Reis for their helpful comments on the paper, and Sebastián Duchêne for help with data analyses and for stimulating discussions.

Additional Information and Declarations

Competing Interests

Author Contributions

The authors declare there are no competing interests.

Martyna Molak conceived and designed the experiments, performed the experiments, analyzed the data, wrote the paper, prepared figures and/or tables, reviewed drafts of the paper.

Simon Y.W. Ho conceived and designed the experiments, wrote the paper, prepared figures and/or tables, reviewed drafts of the paper.

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
