# Peer review of "Prolonged decay of molecular rate estimates for metazoan mitochondrial DNA"

_PeerJ, doi:10.7717/peerj.821_

## Round 0.1 · original submission · Minor Revisions

Both Referees tick the minor revisions box and I agree. To me, the biggest issue is with the clarity with which the methods are described, particularly the randomization element but to a lesser extent throughout. The current text works well for those already in the field but is rather impenetrable for those outside. I would like to see the randomization approach spelled out in detail rather than requiring the reader to go back to the original references. Thus, I was not entirely sure what the issue of self-correlation is in the first place (sorry!), let alone why the randomized relationship had a slope of -1. Elsewhere, I would encourage the authors to read their methods section and try to put themselves in the mindset of a non-specialist, adding a little extra explanatory text wherever this will help make their study accessible to a broader audience. Both Referees make number of other constructive suggestions for improvement.

·

Basic reporting

In my opinion, the article meets all of the relevant standards for the journal.

Experimental design

NA

Validity of the findings

The stated results are supported by the data presented.

Additional comments

Comments on "Prolonged decay of molecular evolutionary rates in metazoan
mitochondrial DNA"
(PeerJ) by Molak and Ho.

This paper looks at systematic biases in evolutionary molecular rate
estimates using the molecular clock. The authors show that the biases
seen previously are general across taxa and extent deeper in time.

Prof. Ho has worked extensively on molecular clocks and is broadly
knowledgable about all aspects. This paper is well written and I feel
that the stated results are supported by the data presented.

I have minor concerns that the timeframes considered here are very
long and the inference of rates over such a long time frame are
difficult. Relaxed clocks, where the rates are indeed likely to
change with time, would be the most appropriate. And intrinsic
differences in rate among taxa should be expected. It could be
explained better how these factors are each accounted for.

I have a minor concern over choosing the oldest calibration point (line
128). Other papers by these same authors have clearly demonstrated that
multiple calibration points work much better. Would adding multiple
points be capable of removing, controlling for or mitigating this
negative relationship?

Line 217: I don't understand how reptiles with a slope of -0.24 and
p=0.014 can be "significant" with a slope of -1.0.

Line 232: The immediately preceeding statements are very precise, while
this one is rather vague.

Line 294: Sentence structure is mangled.

·

Basic reporting

No comment

Experimental design

The randomization test (lines 178-188) needs to be better explained. I really did not understand this part. Do the authors have an explanation about why the slope is -1 for the randomized data?

Validity of the findings

In general I'm happy with the findings.

Additional comments

This paper deals with the apparent change in the molecular evolutionary rate of the mitochondrial genome as a function of the divergence time. This is an interesting topic and the paper gives a meta-analysis of ready available and new data. Overall I think the paper is fine but I have a few comments.

The authors show a conspicuous trend where the molecular rate is negatively correlated with the divergence time, i.e. old branches in the metazoan phylogeny show low mitochondrial rates, while young branches show fast rates. The paper is mostly exploratory and the causes for this are speculated upon. The authors propose that the failure of current substitution models to correct adequately for multiple substitutions is responsible for the apparent negative correlation, i.e. the number of substitutions is underestimated for ancient divergences and hence the underestimation in the molecular rate. In general I agree with this point. We recently studied the pattern of change in omega (=dN/dS) in mitochondrial genomes (dos Reis and Yang, 2013, Genetics), and we proposed model misspecificaiton as the cause for this. Although we did not focus on divergence time estimation on that paper, I think the process is the same. That is, underestimation of non-synonymous substitutions causes an apparent decay in both dN/dS and in the estimated molecular rate with older divergences. The authors may want to look at our mathematical analysis.

Now, if the cause for the apparent decay is model misspecification, then the decay is an artifact, and it may well be the case that mitochondrial rates have remained roughly constant throught the 800My or so of Metazoan evolution (of course we don't know this yet but it seems sensible). In this sense, the title of the paper is strange, as it would suggest that the trend may be real. Maybe something on the lines of "Estimated molecular rates decay with divergence time along the Metazoan phylogeny" or the like. This is just a suggestion.

---

## Round 0.2 · accepted · Accept

I am happy with the changes that have been made. The English is pretty good but could be improved if edited by a native speaker and this would further improve the MS, particularly in the more technical sections.